# Case Study upon Foliar Application of Biofertilizers Affecting Microbial Biomass and Enzyme Activity in Soil and Yield Related Properties of Maize and Wheat Grains

**DOI:** 10.3390/biology9120452

**Published:** 2020-12-08

**Authors:** Dragana Latkovic, Jelena Maksimovic, Zoran Dinic, Radmila Pivic, Aleksandar Stanojkovic, Aleksandra Stanojkovic-Sebic

**Affiliations:** 1Faculty of Agriculture, University of Novi Sad, Trg Dositeja Obradovića 8, 21000 Novi Sad, Serbia; dragana@polj.uns.ac.rs; 2Institute of Soil Science, Teodora Drajzera 7, 11000 Belgrade, Serbia; jelena.maks@yahoo.com (J.M.); soils.dinic@gmail.com (Z.D.); drradmila@pivic.com (R.P.); 3Institute for Animal Husbandry, Autoput 16, P.O. Box 23, 11000 Belgrade-Zemun, Serbia; izs.aleksandar@gmail.com

**Keywords:** crops, dehydrogenase activity, microbial biomass, microbial inoculants, NPK fertilizers, proteinase activity

## Abstract

**Simple Summary:**

The study evaluated the effects of the application of microbial inoculants (N-fixing *Klebsiella planticola* and *Enterobacter* spp.), two rates of composite mineral fertilizers, and their combination on microbial biomass carbon, dehydrogenase and proteinase activity in acid Lessivated Cambisol and yield-related properties of maize and wheat grains in a two-year trial. The results indicated that mineral fertilizers are not, in general, negative for soil microbiota when used in the context of sustainable agriculture without monoculture. It was found that the combined application of microbial inoculants and lower doses of mineral nitrogen, phosphorus, and potassium (NPK) fertilizers, increased the yield of maize and wheat compared to the use of only NPK fertilizers in our case study. The highest values of the examined parameters of soil fertility were determined in the period with better precipitation distribution during the vegetation period of the studied year due to the optimal values of soil moisture and air temperature. Concluding, the authors point out the task of science to offer the producer a solution for conducting intensive agricultural production, which must be economically and ecologically safer.

**Abstract:**

This study evaluated the effects of the application of microbial inoculants (N-fixing *Klebsiella planticola* and *Enterobacter* spp.), two rates of composite mineral fertilizers, and their combination on microbial biomass carbon (MBC), dehydrogenase (DHA), and proteinase activity (PTA) in Lessivated Cambisol and yield-related properties of maize and wheat grains in a two-year trial. Unfertilized soil was used as a control variant. MBC was measured using the chloroform fumigation-extraction method, DHA was determined spectrophotometrically by measuring the intensity of the formed red-colored triphenyl formazan, while PTA was determined using a titration method by measuring the degree of gelatine decomposition. In grain samples, P was determined spectrophotometrically, K—by flame emission photometry, N—on an elemental carbon/nitrogen/sulfur (CNS) analyzer, and crude proteins—by calculation of N content. Measuring both crops’ yield was carried out at the end of the vegetation. The results indicated that mineral fertilizers are not, in general, negative for soil microbiota when used in the context of sustainable agriculture without monoculture. There is a significant increase in the values of soil MBC, DHA, and PTA in the variants with combined application of bacterial inoculants and lower rates of mineral fertilizers. The highest values of these parameters were determined in the period with a better distribution of precipitation during the vegetation period of the year. The mentioned combination also resulted in a higher grain yield of maize and wheat comparing to the application of lower rates of the NPK nutrients solely. The combined application of high rates of mineral fertilizers and bacterial inoculants resulted in significantly increased N, P, K, and protein content in the grains of crops, and the same applied to yield. Concluding, studied bacterial inoculants can be used to specify the replacement of nitrogen fertilizers, stimulating the microbial biomass and enzyme activity in the soil, helping to ensure that the supply of nutrients contributing to an optimized yield of crops is maintained.

## 1. Introduction

Modern crop production, which tends to provide and favors higher yields, involves an intensive application of fertilizers and pesticides, which considerably affects the disturbance of the environmental balance. The persistence in the traditional farming systems opens, directly or indirectly, the possibility of environmental degradation and destruction of soil as the most important resource in the food chain [1].

Since maize and wheat occupy a very important place in crop production, this research has the task to identify ways of the rational use of mineral fertilizers and the possibilities of application of the nitrogen-fixing bacteria with the addition of suitable mineral fertilizer, whereby the biological balance in the soil would not be substantially impaired. On the other hand, it would allow for maintaining or increasing soil productivity, yield quality, and the production of safe food [2]. The incorporation of high doses of mineral fertilizers into soils and their long-term usage does not always have the expected effect. In the beginning, it may come to a sudden intensification of microbial processes in the soil, which can be justified if the primary task is increasing yields. However, the consequences could be various, such as disruption of the biological balance, the loss of mineral fertilizers, loss of physical, chemical, and biological properties of the soil, as well as other environmental disturbances, especially in the cases of mismanagement and not in the case of best mineral practice [2,3,4,5,6].

Mineral fertilizers are not negative for soil microbiota when used in the context of sustainable agriculture without monoculture. The different nitrogen fertilization regimes affected the soil microbiota differentially depending on the soil habitat, although it should be considered that it is not possible to extrapolate general conclusions when only one soil habitat is analyzed [7].

The amounts of nitrogen fertilization should equalize the export by harvest, plus a part of losses, also given by meteorological events. Fertilizers enhance plant growth and the development of plant biomass. Root biomass is enhanced as well as residues amounts at harvest. These higher amounts of organic compounds support the development and activity of microbiota in soil, not only microflora and microfauna but also mesofauna with positive effects on soil structure, so humus formation with chemical stabilization of structure and pore formation [8].

Creating the optimal conditions for growing wheat and maize in order to achieve high yields is primarily related to the rational implementation of nitrogen fertilizers [9]. The reserves of nitrogen in the atmosphere are virtually inexhaustible. However, the molecular form of nitrogen from the air is only accessible to microorganisms. Some of them live in the soil as free-living; others live in the more or less close symbiosis with the plants, while the third actively fix nitrogen from the air in association with other microorganisms. For this reason, the environmental aspects of farming in the world are based on the development of the scientific basis of a wide application of the microbiological fixation of atmospheric nitrogen in order to reduce the consumption of mineral fertilizers [10]. Therefore, the microbial inoculants-based biofertilizers (including bacteria, fungi, and algae) may be an alternative and/or supplement to mineral nutrients in the production of agricultural crops. By their application, the use of expensive nitrogenous fertilizers can be reduced, and the adoption of phosphorus by plants could be facilitated. The dynamics and orientation of microbiological processes could also be directed, leading to the changes in soil fertility [11,12].

The most common free-living nitrogen-fixing bacteria in the rhizosphere of agricultural crops, such as wheat and corn, belong to the family Enterobacteriaceae, in the genera *Klebsiella* and *Enterobacter* [13].

*Klebsiella* species are Gram-negative, facultative anaerobes, and lactose-fermenting bacteria. They naturally occur in the soil, and about 30% of their strains can fix nitrogen in anaerobic conditions. As the free-living nitrogen-fixing bacteria, they are of agricultural interest as it has been demonstrated that soil inoculation with *Klebsiella* strains can increase the crop yields in agricultural conditions [14]. In this respect, the results of Emtsev et al. [15] indicate the importance of the use of *Klebsiella planticola*, TSHA-91 strain, in agricultural production, not only because of their high nitrogenase activity but also of their adhesion capacity, the ability to absorb at the plant roots and colonize them throughout the growing season, and their ability for growth inhibition of some pathogenic fungi. According to other findings [16,17], the ability to fix nitrogen, produce biologically active matter, and increase crop yield is also a characteristic of other species of the *Klebsiella* genus.

*Enterobacter* species are Gram-negative, facultative anaerobes, and non-spore-forming bacteria. Certain *Enterobacter* species have been found to enhance plant growth as they possess multiple growth-promoting activities [18,19]. Accordingly, endophytic *Enterobacter* free-living species with plant growth-promoting traits have been isolated from the rhizosphere of maize [20] and wheat [13].

In addition, some strains of these bacterial species are also well-known plant growth simulators due to the production of indole-3-acetic acid (IAA) and the solubilization of mineral phosphates.

The capacity of *Klebsiella planticola* strain TSKhA-91 to synthesize indole-3-acetic acid (IAA), other auxins, and its effect on the development of cucumber (*Cucumis sativus* L.) seeds were studied by Blinkov et al. [21]. Accordingly, under conditions of low-temperature stress, the protective effect of *K. planticola* TSKhA-91 on the development of cucumber seeds and stimulation of germination and root formation by its seeds were shown. Similarly, Borham et al. [22] were evaluated phosphate solubilization effects of *Enterobacter cloacae* (strain B1) through its impact on growth and yield characteristics of wheat, cultivar Masr 1, in comparison with chemical fertilizers (phosphoric acid, calcium superphosphate, and NPK), under field conditions. Results of their study showed that *E. cloacae* (B1) effectively increased growth characteristics including plant height, fresh weight, dry weight, flag leaf area, and physiological characteristics including chlorophyll pigments content and yield characteristics such as spike length, spikelets number, grains number/spike, 1000 kernels weight, spike weight, biological weight and grains weight, in comparison with control treatment. Opposite to this, the results of Widowati et al. [23] showed that the germination percentage and rate of soybeans were not significantly different between controls and *Enterobacter cloacea* isolate H3 treated samples.

In the context of the total microbiological activity in soil, important soil biogenetic factors are microbial biomass and dehydrogenase and proteinase activity. Microbial biomass (bacteria, fungi, and Actinomycetes) represents the mass of “live components” of the organic matter in the soil, and their determination allows for a better overview of the reaction of soil microflora to seasonal changes, applied agro-technical measures (tillage and fertilization), and the presence of soil pollutants [24,25]. Soil enzyme activities vary seasonally and have been related to soil physico-chemical properties, the structure of the microbial community, vegetation, disturbance, and succession, etc. [26]. Among all enzymes in the soil environment, soil dehydrogenases (EC 1.1.1.) are one of the most important since they are used as an indicator of an overall soil microbiological activity [27]. In the process of respiration, dehydrogenases catalyze the reaction of the hydrogen separation from a donor, such as various organic compounds, including carbohydrates, organic acids, amino acids, alcohols, humic acids, etc., to the acceptor. Higher dehydrogenase activity indicates a higher respiration potential, i.e., the more intense mineralization of organic matter and humus [28,29]. Although the activity of soil proteases (EC 3.4.4.) does not very much depend on the microbial community of soils, it expresses overall soil biological activity, and it is associated with soil fertility by regulating the amount of available nitrogen in plant and plant growth. Soil proteases are an important factor in nitrogen mineralization, more active in soils with high water and humus content. Nevertheless, the processes of transformation of organic forms of nitrogen are enabled by the effect of microbiological proteases contained in a high number of bacteria, fungi, and Actinomycetes. Peptides and amino acids develop in the process to be further mineralized to the ammonium form and used in plant and microorganism nutrition [26]. An affecting character of fertilizers on enzyme activity and microbial biomass in soil is greatly determined by the multiple-effect of various environmental factors, as follows: soil moisture, pH, oxygen regime, temperature, level of cultivation, and crop species [2], as well as living organisms such as microfauna and mesofauna. Adoption of non-sustainable agricultural practices reduces soil biodiversity [30,31], and examples include the negative influence of tillage, mineral fertilizers, and pesticides on the genetic, functional, and metabolic diversity of soil microorganisms [32,33,34,35]. It is known that animals living in the soil and upper soil can be viewed as facilitators of bacterial and fungal activity and diversity since most soil processes and biogeochemical cycles are regulated not only by soil microbial communities but by the whole soil–food web [36], although it is not discussed in our case study.

Having in mind the above mentioned, the aim of this investigation was to examine the influence of selected soil bacterial inoculants (N-fixing *Klebsiella planticola* and *Enterobacter* spp.), two rates of mineral fertilizers (composite NPK (15:15:15)), and their combination, as well as sampling period on microbial biomass carbon, dehydrogenase and proteinase activity in acid Lessivated Cambisol and yield-related properties of maize and wheat grains in a two-year trial.

## 2. Material and Methods

### 2.1. Field Trial and Weather Conditions

The trial was conducted in the experimental field of the Institute of Soil Science in Mladenovac town, located about 55 km from Belgrade, Serbia (grid reference: 44°24′ N, 20°40′ E). The field was established in 1961, primarily in order to perform long-term field trials and monitor the effects of fertilization with different types and rates of mineral fertilizers on the changes of soil properties and yield of wheat and maize. Within this experimental field in the fall of 2012, an additional trial, which included microbial inoculation of the soil and plants (in 2012/2013 for wheat, in 2014 for maize), was performed.

The studied soil type was Lessivated Cambisol [37]. The experiment was set up in a randomized block design with three replicates on experimental plots of 60 m^2^ (10 × 6 m), based on the following treatments: control (Ø, non-fertilized soil); 60 kg ha^−1^ N and P_2_O_5_, and 40 kg K_2_O ha^−1^ (N1); 120 kg ha^−1^ N, P_2_O_5_ and K_2_O (N2); *Enterobacter* spp. strains + 60 kg ha^−1^ N and P_2_O_5_, and 40 kg K_2_O ha^−1^ (ES + N1); *Enterobacter* spp. strains + 120 kg ha^−1^ N, P_2_O_5_ and K_2_O (ES + N2); *Klebsiella planticola* + 60 kg ha^−1^ N and P_2_O_5_, and 40 kg K_2_O ha^−1^ (KP + N1); *K*. *planticola* + 120 kg ha^−1^ N, P_2_O_5_ and K_2_O (KP + N2). Winter wheat (cv. Evropa 90) and maize (hybrid ZP-341, FAO 300) were included in the trial and sown in optimal terms (first week of November in 2012 for wheat and the third week of April in 2014 for maize).

The relation between wet and dry periods during 2013, 2014, and the months of October, November, and December in 2012, for the study locality is presented using climate diagram according to Walter (Figure 1). This relation was relatively favorable for agricultural production during 2013, which referred a priori to a good distribution of precipitation in the period of March–June, except in April, when it was measured approximately 27.0 mm of precipitation. In the period of October–December 2013, the distribution of precipitation was relatively favorable. The year 2014 was extremely wet, as the highest registered precipitation sum was approximately 240.0 mm, measured during May, and the lowest was about 14 mm (February) and 16.5 mm (November). The relation between wet and dry periods during 2014 was extremely unfavorable for agricultural production.

In general, monthly air temperature did not differ noticeably among studied years. However, the registered total precipitation sum was considerably much higher in 2014 than in 2013 (Figure 1), since 2014 was a year with great floods registered across almost the whole territory of the Republic of Serbia.

### 2.2. Mineral Fertilization

Application of the composite NPK mineral fertilizer was carried out as follows: nitrogen (N) fertilizer was applied in the form of urea with 46% N; phosphorus (P) fertilizer was applied in the form of monoammonium phosphate with 52% P_2_O_5_ and 11% N, and potassium (K) fertilizer as a 40% potassium salt (potassium chloride, KCl).

Fertilization of each experimental plot was done manually. The established amount of fertilizer for maize was applied in the spring in 2014, before sowing, while for wheat phosphorus, potassium, and one-third of the nitrogen fertilizer were applied in fall 2012 before sowing, and the remaining two-thirds of the nitrogen fertilizer was applied in March the following year (2013), at the tillering stage.

### 2.3. Bacterial Inoculation

The pure cultures of *Enterobacter* strains (KG-75 and KG-76) were obtained from the collection of microorganisms of the Microbiology Laboratory in the Centre for Small Grains (Kragujevac, Serbia), where they have been isolated from the rhizosphere of wheat. N-fixing gram-negative bacterium *K*. *planticola* (strain TSHA-91) was directly obtained from the stock culture of the Microbiology Laboratory of Faculty of Agronomy (Čačak, Serbia), and indirectly from the collection of microorganisms of the Laboratory of Microbiology, Moscow Timiryazev Agricultural Academy (Russia). The solid nutrient medium for *K. planticola* consisted of the following chemical compounds: peptone 1: 1.20 g; K_2_HPO_4_: 0.50 g; KH_2_PO_4_: 0.30 g; MgSO_4_: 0.10 g; CaCl_2_: 0.03 g; sucrose: 6.00 g; (NH_4_)_2_SO_4_: 0.14 g; yeast extract: 0.10 g; agar: 16.00 g; distilled deionised water: 1000 cm^3^; pH 7.3. Both strains of *Enterobacter* were cultivated on the solid meat-peptone agar MPA (Torlak, Belgrade) with the following chemical composition: peptone 1: 15.00 g; meat extract: 3.00 g; NaCl: 5.00 g; K_2_HPO_4_: 0.30 g; agar: 18.00 g; distilled deionized water: 1000 cm^3^; pH 7.3.

Liquid concentrated inoculums of the tested N-fixing bacteria for upper soil and plant inoculation were obtained by transferring them from a solid nutrient medium to 250.0 cm^3^ of an appropriate liquid nutrient medium (2 × 250.0 cm^3^ for *Enterobacter* strains, 2 × 250.0 cm^3^ for *K. planticola*), followed by incubation for 24 hours in a thermostat with a shaker at 28 °C ± 1. The liquid nutrient media were of the same chemical composition as the solid but without agar.

After incubation, the inoculums of joint *Enterobacter* strains KG-75 and KG-76 in an amount of 500 cm^3^ (2 × 250.0 cm^3^) and an inoculum of *K. planticola* in an amount of 500 cm^3^ (2 × 250.0 cm^3^) were transferred to fermenters with an appropriate liquid nutrient medium in an amount of 18,000.00 cm^3^ each and incubated in a thermostat with aeration for 48 hours at 28 °C ± 1.

The liquid inoculum for upper soil and plants treatment was made by diluting 18,000.0 cm^3^ of obtained concentrated liquid inoculum of joint *Enterobacter* strains (100–180 × 10^7^ cells per 1.0 cm^3^ of inoculum) and 18,000.0 cm^3^ of obtained concentrated liquid inoculum of *K. planticola* (100–300 × 10^7^ cells per 1.0 cm^3^ of inoculum), respectively, in 32,000.0 cm^3^ of the tap water under field conditions, using containers of 60,000.0 cm^3^.

The first upper soil and foliar sprinkling (basic fertilization) with made bacterial inoculums were carried out during the stage when both crops had two–three formed leaves, using a plastic haversack sprinkler with 300.00 cm^3^ m^−2^ of diluted liquid bacterial inoculum.

The second upper soil and the foliar sprinkling of maize with bacterial inoculums were carried out at the stage, at nine–ten formed leaves, and of wheat at the tillering stage, a few days after fertilizing with mineral nitrogen. The titer of the 18,000.0 cm^3^ concentrated liquid inoculum of *K. planticola* was 50–100 × 10^7^ cells per 1.0 cm^3^ of inoculum, and of 18,000.0 cm^3^ concentrated liquid inoculum of joint *Enterobacter* strains: 75–175 × 10^7^ cells per 1.0 cm^3^ of inoculum. The procedure for the preparation of bacterial inoculants for the second soil and foliar sprinkling was the same as for basic fertilization.

However, roots were not sprinkled with bacterial inoculums directly, only in the zone of upper soil, about 50 cm in diameter around the plants. All aboveground parts of maize and wheat were subjected to foliar sprinkling with bacterial inoculums.

### 2.4. Soil Analysis

The preliminary observation of the upper soil included an adequate sampling from the depth of 0–20 cm, with three replicates per experimental plot of 60 m^2^ (10 × 6 m), and preparation, where the soil samples were air- and absolutely-dried, crushed and passed through a sieve with a diameter of ≤2 mm [38], followed by chemical analyses and granulometric composition. The following chemical parameters were analyzed: soil acidity (pH in H_2_O and 1M KCl, *v*/*v*: soil:H_2_O = 1:5, soil:1M KCl = 1:5) was analyzed potentiometrically, using glass electrode [39]; total nitrogen content (N) was analyzed by dry combustion using elemental CNS analyzer Vario EL III [40]; available phosphorus (P) and potassium (K) were analyzed by AL-method according to Egner-Riehm [41], where K was determined by flame emission photometry and P by spectrophotometer after color development with ammonium molybdate and stannous chloride; organic carbon content (Corg) was determined after dry combustion on elemental CNS analyzer, Vario model EL III [42]. The share of clay, sand, and silt fractions in soil was analyzed by determination of particle size distribution in mineral soil material, using a standardized method by sieving and sedimentation [43], after which the textural soil class was determined using the International Union of Soil Science (IUSS) texture triangle [44].

Collection, handling, and storage of upper soil without plant roots samples under aerobic conditions for microbiological analyses were carried according to SRPS ISO 10381–6:2000 method [45]. The soil was sampled from the plow layer (0–20 cm) two times during the vegetation period of wheat (the beginning of tillering—sampling period I, full-grain maturity stage—sampling period II), and three times during the vegetation period of maize (intensive plant growth stage, seven–eight leaves—sampling period I, milk-waxy maturity stage—sampling period II, full-grain maturity stage—sampling period III). The sampling period was factor B in statistical data analysis.

Soil microbial biomass in the form of CO_2_-C (microbial biomass carbon—MBC) was measured using the chloroform fumigation-extraction method on the basis of differences in the amount of segregated CO_2_ between fumigated and non-fumigated soil [46]. The MBC was calculated using the equation MBC = (Fc − UFc)/Kc, where MBC is the microbial biomass carbon, Fc is CO_2_ in fumigated soil samples, UFc is CO_2_ in non-fumigated soil samples, and Kc is the conversion coefficient of C to CO_2_, rating 0.45 for most soils. The obtained value of MBC was at the end expressed in mg kg^−1^ of absolutely-dry soil.

Soil dehydrogenase activity (DHA) was assayed under standard conditions (24 hours of incubation at 30 °C ± 1) by measuring the intensity of the red-colored triphenyl formazan (TPF) extinction, formed by reduction of 2,3,5-triphenyltetrazolium chloride, spectrophotometrically [47]. The intensity of the developed color was immediately counted on a spectrophotometer at a wavelength of 546 µm. The quantity of TPF was read from a standard curve with concentrations of 0.25, 0.5, 1.0, 1.5, and 2.0 mg of TPF. The obtained value of DHA was at the end expressed in µg TPF 10 g^−1^ of an air-dry soil.

The method for the determination of the soil proteinase activity was based on measuring the degree of decomposition (hydrolysis) of gelatine, using a titration method, according to Romejko [48]. The titration was carried with 4% ferric chloride hexahydrate, with the addition of two to three drops of the solution of ammonium thiocyanate indicator until the orange color occurred. Proteinase activity (PTA) was determined based on the amount of ferric chloride hexahydrate used in the titration. The obtained value was at the end expressed in the number of gelatinolytic units g^−1^ of an air-dry soil, wherein 0.2 cm^3^ of titration mean corresponded to ten gelatinolytic units.

### 2.5. Grain Chemical Traits and Yield

The harvesting and measuring of the grain yield of maize and wheat from each plot were carried out at the end of the vegetation, at the full-grain maturity stage. Wheat was harvested in the third week of June in 2013, while the harvesting of maize was done in the first week of October in 2014. The data on grain yield was adjusted to 14% moisture content and calculated into t ha^-1^.

After harvesting, the grain samples were taken [49] and weighed before and after drying at 105 °C. The chemical analyses of the grains included the determination of phosphorus (P) and potassium (K) contents, using “wet” combustion, i.e., the grains were heated to boiling with the mixture of concentrated sulphuric and perchloric acids. In the obtained solution, P was determined spectrophotometrically with molybdate [50], and K—by flame emission photometry [51]. An elemental CNS analyzer, Vario model EL III, was used in the determination of nitrogen (N) content [52], on the basis of which the content of the crude proteins in dry matter was calculated, using the calculations suggested by FAO [53]: crude proteins (%) = N (%) × 6.25 (factor for conversion of N content to crude protein in maize grain), crude proteins (%) = N (%) × 5.83 (factor for conversion of N content to crude protein in wheat grain), respectively.

### 2.6. Statistical Analysis

The obtained data on chemical parameters from soil materials are presented as arithmetic means of three replicates, standard deviation values, and range, while the data on soil microbial biomass and enzyme activity was analyzed using an analysis of variance (ANOVA) test, using SPS Statistica Software. The significance of the differences between the study factors was compared by the least significant difference (LSD) test at *p* < 0.05 and *p* < 0.01. The effects of applied fertilizers on the yield and chemical composition of maize and wheat grain were evaluated using an ANOVA test (SPSS 20.0, Chicago, IL, USA), followed by Duncan’s Multiple Range Test (DMRT). Significant differences between means were tested by the LSD test at *p* = 0.05 and *p* = 0.01.

## 3. Results and Discussion

### 3.1. Soil Chemical and Physical Properties

The studied surface of Lessivated Cambisol is characterized by very acid reaction, high available potassium, medium to high available phosphorus, and medium total nitrogen and organic carbon supply. According to granulometric composition, Lessivated Cambisol is a clay loam and has a relatively favorable particle size distribution for the cultivation of wheat and maize (Table 1).

### 3.2. Fertilization Effect on Soil Microbial Biomass, Dehydrogenase and Proteinase Activity

Experimental data on the studied fertilization treatments effects on the average values of the microbial biomass carbon (MBC), dehydrogenase (DHA), and proteinase (PTA) activity in soil under winter wheat and maize, observed in two vegetation periods, are given in Table 2, Table 3, Table 4, Table 5, Table 6 and Table 7. The highest and statistically highly significant (*p* < 0.01) level of MBC and enzyme activity (DHA and PTA) inhibition in the upper soil was determined in the treatment with high rates of NPK nutrients (N2 variant) during both studied vegetation periods of wheat and maize. Opposite to this, the highest and statistically highly significant (*p* < 0.01) stimulation of the soil MBC and DHA in soil under wheat, and MBC and PTA in soil under maize, was affected by the treatments which included a combination of the low rates of NPK fertilizers and bacterial inoculants used (variants ES + N1 and KP + N1). Similarly, these treatments caused a statistically significant (*p* < 0.05) stimulation of PTA in soil under wheat, although the variant ES + N1 affected statistically significant (*p* < 0.05), and the variant KP + N1 statistically insignificant (*p* > 0.05) on DHA in soil under maize. Different treatments were factor A in the statistical data analysis.

Statistical data analysis determined the existence of statistical significance of the effect of sampling period studied (factor B) for all observed parameters in soil under wheat, as well as the statistical significance (*p* < 0.05) for PTA and lack of statistical significance (*p* > 0.05) for MBC and DHA in soil under maize. Higher and statistically highly significant (*p* < 0.01) values of MBC in soil under wheat were determined in the first period of sampling compared to the second. Similar results were obtained for soil DHA under wheat, although the values were statistically significant (*p* < 0.05) compared to the second period of sampling. Opposite to the values of MBC and DHA, higher values of PTA in soil under wheat were determined in the second sampling period, which was statistically significant (*p* < 0.05) compared to the first period of sampling. As for maize, the highest values of all parameters were observed in the second vegetation period of maize and the lowest in the first period of sampling, when extremely high precipitation was registered.

The interaction effect of fertilization treatment and sampling period (A × B) indicated insignificancy in the effect of the fertilizers applied depending on the studied vegetation period of both crops studied for MBC, DHA, and PTA (Table 2, Table 3, Table 4, Table 5, Table 6 and Table 7).

Biological productivity of various types of soil is defined by the composition, structure, and activity of their biological component, in which most biological processes occur thanks to the enzyme systems of microorganisms, accounting for 60–90% of the total metabolic activity in soil. Also, the interactions between microorganisms, microfauna, mesofauna, and macrofauna are of key importance. According to Culliney [54], macrofaunal organisms are ecosystem engineers able to ameliorate soil physical structure, mineral and organic matter composition, and hydrology, influencing nutrient and energy flow and forming a connection between the food chains of the foliage and the soil.

The diversity of microorganisms in the regulation of biological fertility and plant nutrition is the primary exploitation of agricultural soil. However, the disruption of biological balance often occurs after long-term usage of high rates of mineral fertilizer use [55]. Kanchikerimath and Singh [56] found that the balanced use of mineral fertilizers affects the increase in enzyme activity in the soil. The long-term application of the high rates of mineral fertilizers significantly lowers the soil pH [57] and thus affects the decrease in microbial biomass and enzyme activity [27]. These findings are in accordance with the present case study, where the values of microbial biomass, dehydrogenase, and proteinase activity in soil depended significantly on the fertilizers rates applied, whereby high doses of mineral fertilizers significantly decreased these biological parameters in soil.

The knowledge about the dominance of beneficial microorganisms to direct the processes in soil towards increase the soil fertility led to a number of studies, which referred to the application of useful microorganisms in crop production [12,58,59]. In a study by Araújo et al. [60], the application of biofertilizers that contained beneficial microorganisms increased soil microbial biomass carbon and soil respiration as compared with controls.

By studying the effect of two rates of mineral nitrogen application and *Klebsiella planticola* SL09-based microbial biofertilizer on the parameters of soil biogenity in the cultivation of potato, Mandić et al. [2] determined an increase in microbiological activity and potato yield in variants where *K. planticola* was used. Similarly, the present study also emphasized the higher activity of microorganisms in the soils treated with the combination of the associative microbial inoculants and low rates of mineral fertilizers. Stimulative effects of the combined usage of the associative N-fixing bacteria and low rates of NPK nutrients on soil microbial biomass and enzyme activity were also reported in the study of Cvijanović et al. [61]. The data of Oliveira et al. [59] highlighted the feasibility of partially substituting the N-fertilizer demand in non-legume crops using high-quality inoculants formulations prepared with N-fixing bacteria.

The significance of the influence of the sampling period on soil microbial biomass and enzyme activity was reported in our case study. The highest values of the studied parameters of soil fertility were determined in the period with a better distribution of precipitation during the vegetation period of the year studied, probably due to the optimal values of the soil moisture and air temperature. Similar to this, Nagaraja et al. [62] reported an increase in biological activity with an optimal increase in soil moisture status during the wet periods of the year.

### 3.3. Fertilization Effect on Yield and Chemical Composition of Wheat and Maize Grain

By analyzing the yield and grain chemical composition of both crops tested, including nitrogen, phosphorus, potassium, and proteins during the two-year study, it was determined the significant differences between these parameters and applied fertilization treatments (Table 8 and Table 9). The treatment, which included the combination of high rates of mineral NPK fertilizers and microbial inoculants used (variants ES + N2 and KP + N2), as well as the treatment with high rates of mineral NPK fertilizers (variant N2), brought about a statistically significant increase in the share of nitrogen, phosphorus, potassium, and protein in the grains of both crops studied compared to the other tested variants. Similarly to this, the combined use of microbial inoculants and high rates of mineral NPK fertilizers (variants ES + N2 and KP + N2) caused an increase in yields of maize and wheat in our case study, which is the main connection with mineral nutrition. According to Szeuczuk de Oliveira et al. [63], there is a correlation between the nitrogen and phosphorus levels in grain and between phosphorus levels and grain yield.

Similar results were obtained in a study by Dasci and Comakli [64], in which the crude protein content in forage maize from fertilizers application variants was higher than that of the plots without fertilization. Nevertheless, the present results on fertilization impact on yield and chemical composition of crops grain are not in accordance with the obtained data on soil microbial biomass and enzyme activity. Decreased enzyme activity and microbial biomass in this study are most likely the result of a long-term mineral fertilizers introduction in the soil at high rates, bringing about a change in its agrochemical properties. On the other hand, this did not negatively affect the mineral nutrition and yield of maize and wheat cultivated on studied soil type and existing climatic conditions.

A different result can be found in the study of Souza et al. [65], in which it was determined the negative effects of an application of higher doses of nitrogen at sowing following inoculation with *Azospirillum brasilense* on the accumulation of N in shoots, roots, grains and shoot dry mass of maize; thus, reducing the maize grain yield. Opposite to this, Dalla Santa et al. [66] noted a significantly higher yield of maize in variants that had been treated with microbiological fertilizer and high rates of mineral nitrogen in the amount of 150 kg ha^−1^. Additionally, in a study by Teixeira Filho et al. [67], an increment in mineral N rates in association with *Azospirillum brasilense* inoculation increased the N wheat grain concentration.

The hypothetical excess of fixed microbiological nitrogen, obtained by potential increased activity of applied microbial inoculants, with lower amounts of mineral nitrogen (variants ES + N1 and KP + N1), also may influence positively on the yield of maize and wheat grain, which was confirmed in the study of Bhardwaj et al. [10]. Nevertheless, the combined application of the microbial inoculants and low rates of mineral NPK fertilizers (variants ES + N1 and KP + N1) caused an increase in yields of both crops compared to the application of low rates of mineral NPK fertilizers in conditions of agricultural production typical for this case study.

In the study performed by Bákonyi et al. [68], the applied living bacteria containing biofertilizer stimulated the germination and dry matter production of maize seedlings by reason of excreting phytohormones and enhancing the nutrient mobilization from the seed.

Application of the nitrogen-fixing microorganisms in plant production, in particular in combination with lower amounts of mineral nitrogen, can increase a yield and significantly reduces the use of mineral fertilizers, which indicates the possibility of replacing a certain amount of mineral nitrogen with highly-effective microorganisms [10], especially in the cases of mismanagement and not in the case of best mineral practice. The combined application of the microbial inoculants and low rates of mineral NPK fertilizers has caused an increase in yields of maize and wheat comparing to the application of low rates of mineral NPK fertilizers in our case study. In a study by Oliveira et al. [59], inoculated plants of maize grown under 80% reduction in N fertilizer showed yields at levels compared to fully fertilized plants. In field experiments, Fukami et al. [58] found that inoculation of maize with *Azospirillum brasilense* allowed for a 25% reduction in the need for N fertilizers. The reports of Cassán and Diaz-Zorita [11] regarding the inoculation of dryland crops with *Azospirillum* sp. showed positive grain yield responses on winter (14.0% increase) and summer cereals (9.5% increase).

The potential IAA secretion and P solubilization to promote plant growth and development is a point of ongoing discussion in the literature. 

Gupta et al. [69] reported that plant growth-promoting rhizobacteria (PGPR) are a well-known group of microorganisms able to promote plant growth through enhanced biological nitrogen fixation (BNF), synthesis of plant hormones, soil nutrient solubilization (as phosphorus (P) and potassium (K).

By studying wheat plants, Kumar et al. [70] showed that inoculation with *Bacillus megaterium*, *Arthrobacter chlorophenolicus*, and *Enterobacter* improves grain yield and the amount of P in the straw and grain up to two-fold in greenhouse and field experiments. They also reported that inoculation of efficient P-solubilizer bacteria significantly improve P absorption by plants.

By evaluating nutrient use efficiency and nutrient uptake promoting of rice by potassium solubilizing bacteria, Yaghoubi Khanghahi et al. [71] reported that NPK chemical fertilizer treatment showed better results than other treatments on K uptake by grain and straw. This result was confirmed by Bakhshandeh et al. [72]. Similarly, the results obtained from the pot and field experiments proved that N and P efficiency was affected by the application of NPK fertilizer and NP½K + potassium solubilizing bacteria (KSB) treatments. However, the results indicated that the bioinoculation with potassium solubilizing bacteria strains isolated from soil paddies could be considered as an effective way to increase potassium, nitrogen, and phosphorus uptake by rice plants and enhance their use efficiency and remobilization to grains in flooding irrigation conditions [71].

If we compare the uptake of these nutrients by wheat and corn grain, it is noticeable that the application of *K*. *planticola* and *Enterobacter* spp. was increased at a similar level to the total amount of up-taken P like N and noticeably more of K in comparison with NPK alone independently from the level of N fertilization.

The work of Mendoza-Arroyo et al. [73] has demonstrated that *Enterobacter* sp. ITCB-09 can solubilize inorganic phosphate in higher concentrations than those reported in other bacteria. The strain produces siderophores and extracellular polymeric substances (EPS), which are biomolecules that could contribute to plant growth, control of pathogens, or soil structure, health, and fertility, making this bacterium to have biofertilization potential. Mahmood et al. [74] stated that EPS-producing bacteria could promote plant growth since EPS production in the rhizosphere can increase water availability and nutrients such as phosphorus and potassium, as well as helping plants tolerate salinity.

The interrelationships of the MBC and enzymatic activity with respect to the efficacy of NPK mixed with tested PGPB are discussed in the study of Lavakush et al. [75]. They observed potential in rice plants for reducing the P-fertilization up to 50% once plants received half of the recommended P-fertilizer dose. The plants were inoculated with the P-solubilizing bacteria *Azotobacter chroococcum*, *Azospirillum brasilense*, and combined *Pseudomonas* spp. culture. Inoculated rice plants presented a similar performance benefit in plant height, panicle length, grain number per panicle, and grain yield when fertilized with 30 and 60 kg P ha^−^^1^ in a greenhouse experiment. Similarly, Dutta and Bandyopadhyay [76] showed that a reduction of up to one-third in P-fertilization of chickpea plants (inoculated with P-solubilizing *Pseudomonas* sp.) did not cause any decrease in plant development parameters.

## 4. Conclusions

The obtained results of the conducted research showed that mineral fertilizing with good management is not an obligation but allows for better timing in the course of plant development.

The highest values of the examined parameters of soil fertility were determined in the period with better precipitation distribution during the vegetation period of the studied year due to the optimal values of soil moisture and air temperature.

The effect of fertilization on soil microbial biomass, dehydrogenase, and proteinase activity showed that the level of MBC and inhibition of enzyme activity (DHA and PTA) in upper soil was determined in the treatment with high rates of NPK nutrients (N2 variant) during both wheat and maize vegetation periods. In contrast, the greatest and statistically significant stimulation of MBC and DHA in wheat-grown soil, and MBC and PTA in maize-grown soil, was affected by treatments that included a combination of low rates NPK and bacterial inoculants.

On the upper soil on which wheat was grown, the statistical analysis of data determined the existence of statistical significance of the effect of the examined sampling period for all observed parameters, while there was no statistical significance for MBC and DHA in soil on which maize was grown. Higher and statistically very significant MBC values in soil under wheat were determined in the first sampling period compared to the second. Similar results were obtained for DHA soils under wheat. In contrast to MBC and DHA values, in the second sampling period, higher values of PTA in soil under wheat were found. As for maize, the highest values of all parameters were observed in the second vegetation period, and the lowest in the first sampling period, when extremely high amounts of precipitation were registered.

The effect of the interaction of fertilizer treatment and sampling period indicated the insignificance of the effect of applied fertilizers depending on the examined vegetation period of both cultures studied for MBC, DHA, and PTA.

Analysis of grain yield and chemical composition of both tested cultures, including nitrogen, phosphorus, potassium, and proteins, during a two-year study revealed significant differences between these parameters and the applied fertilization treatments. The best results were obtained in the treatment, which included a combination of high amounts of mineral NPK fertilizers and microbial inoculants. It was also found that the combined application of microbial inoculants and lower doses of mineral NPK fertilizers increased the yield of maize and wheat compared to the use of only NPK fertilizers in our case study.

## Figures and Tables

**Figure 1 biology-09-00452-f001:**
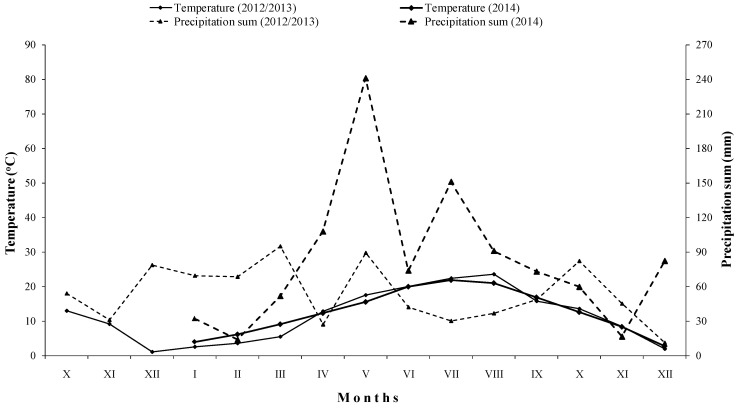
Temperatures and precipitations developments for wheat and maize vegetation period in 2012/2013 and 2014, respectively, for the field study locality.

**Table 1 biology-09-00452-t001:** Main chemical and physical properties of soil materials (average ± standard deviation value *).

Textural Class	Clay Loam
Granulometric composition	
Sand (%), fractions > 0.02 mm	31.8 ± 1.87 *
Clay (%), fractions < 0.002 mm	31.0 ± 1.53
Silt (%), fractions 0.02–0.002 mm	37.2 ± 1.76
Chemical properties	
pH in H_2_O	4.90 ± 0.03
pH in 1M KCl	4.06 ± 0.05
Total N (g kg^−1^)	1.36 ± 0.005
Corg (g kg^−1^)	21.9 ± 0.01
Available P (mg kg^−1^)	157.3 ± 0.31
Available K (mg kg^−1^)	253.0 ± 0.30

**Table 2 biology-09-00452-t002:** Effect of the fertilization treatment (A), sampling period (B), and fertilization treatment and sampling period interaction (A × B) on average MBC (mg kg^−1^ of absolutely-dry soil) in Lessivated Cambisol under wheat.

Tested Parameters								
Ferilization treatment (A)	Control non-fertilized (Ø)	N1 (NPK 60:60:40)	N2 (NPK 120:120:120)	N1 + KP	N2 + KP	N1 + ES	N2 + ES	$\bar{X}$B
First sampling period (B)	84.12	76.87	49.66	128.08	60.77	140.64	65.12	86.47
Second sampling period (B)	56.55	47.54	29.34	82.97	36.43	100.52	38.68	56.00
$\bar{X}$A	70.34	62.21	39.50	105.53	48.60	120.58	51.90	71.24
Statistical analyses	Source of variation							
	Fertilization treatment (A)		Sampling period (B)		Fertilization treatment and sampling period interaction (A × B)			
LSD (0.05)	13.25 ***		6.23 ***		18.32 ^NSD^			
LSD (0.01)	17.37 ***		7.99 ***		24.15 ^NSD^			

LSD—least significant differences at *p* = 0.05 and *p* = 0.01; and *** indicates statistical significant differences; ^NSD^ indicates no significant difference.

**Table 3 biology-09-00452-t003:** Effect of the fertilization treatment (A), sampling period (B), and fertilization treatment and sampling period interaction (A × B) on average DHA (µg TPF 10 g^−1^ of air-dry soil) in Lessivated Cambisol under wheat.

Tested Parameters								
Ferilization treatment (A)	Control non-fertilized (Ø)	N1 (NPK 60:60:40)	N2 (NPK 120:120:120)	N1 + KP	N2 + KP	N1 + ES	N2 + ES	$\bar{X}$B
First sampling period (B)	106.31	102.94	55.74	143.18	89.84	152.18	84.18	104.91
Second sampling period (B)	102.01	100.61	46.84	129.11	71.84	127.44	73.38	93.03
$\bar{X}$A	104.16	101.78	51.29	136.15	80.84	139.81	78.78	98.97
Statistical analyses	Source of variation							
	Fertilization treatment (A)		Sampling period (B)		Fertilization treatment and sampling period interaction (A × B)			
LSD (0.05)	11.28 ***		5.23 **		17.94 ^NSD^			
LSD (0.01)	15.25 ***		7.99 **		23.22 ^NSD^			

DHA: dehydrogenase activity; TFP: triphenyl formazan; LSD: least significant differences at *p* = 0.05 and *p* = 0.01; ** and *** indicates statistical significant differences; ^NSD^ indicates no significant difference.

**Table 4 biology-09-00452-t004:** Effect of the fertilization treatment (A), sampling period (B), and fertilization treatment and sampling period interaction (A × B) on average PTA (number of gelatinolytic units g^−1^ of air-dry soil) in Lessivated Cambisol under wheat.

Tested Parameters								
Ferilization treatment (A)	Control non-fertilized (Ø)	N1 (NPK 60:60:40)	N2 (NPK 120:120:120)	N1 + KP	N2 + KP	N1 + ES	N2 + ES	$\bar{X}$B
First sampling period (B)	10.00	10.00	1.10	18.30	4.50	18.40	5.60	9.70
Second sampling period (B)	11.70	10.60	1.70	20.60	7.30	21.10	6.70	11.39
$\bar{X}$A	10.85	10.30	1.40	19.45	5.90	19.75	6.15	10.54
Statistical analyses	Source of variation							
	Fertilization treatment (A)		Sampling period (B)		Fertilization treatment and sampling period interaction (A × B)			
LSD (0.05)	2.94 **		1.25 **		4.15 ^NSD^			
LSD (0.01)	3.93 ^NSD^		1.67 ^NSD^		5.55 ^NSD^			

PTA: proteinase activity; LSD: least significant differences at *p* = 0.05 and *p* = 0.01; ** and indicates statistical significant differences; ^NSD^ indicates no significant difference.

**Table 5 biology-09-00452-t005:** Effect of the fertilization treatment (A), sampling period (B), and fertilization treatment and sampling period interaction (A × B) on average MBC (mg kg^−1^ of absolutely-dry soil) in Lessivated Cambisol under maize.

	Tested Parameters							
Ferilization treatment (A)	Control non-fertilized (Ø)	N1 (NPK 60:60:40)	N2 (NPK 120:120:120)	N1 + KP	N2 + KP	N1 + ES	N2 + ES	$\bar{X}$B
First sampling period (B)	38.98	21.06	14.84	69.32	19.32	80.66	18.16	37.48
Second sampling period (B)	54.36	37.13	17.20	75.89	23.34	89.25	25.37	46.08
Third sampling period (B)	51.84	36.62	15.01	71.17	21.17	81.00	22.51	42.76
$\bar{X}$A	48.39	31.60	15.68	72.13	21.28	83.64	22.01	42.10
Statistical analyses	Source of variation							
	Fertilization treatment (A)		Sampling period (B)		Fertilization treatment and sampling period interaction (A × B)			
LSD (0.05)	10.64 ***		5.07 ^NSD^		19.17 ^NSD^			
LSD (0.01)	14.49 ***		7.08 ^NSD^		25.83 ^NSD^			

MBC: microbial biomass carbon; LSD: least significant differences at *p* = 0.05 and *p* = 0.01; and *** indicates statistical significant differences; ^NSD^ indicates no significant difference.

**Table 6 biology-09-00452-t006:** Effect of the fertilization treatment (A), sampling period (B), and fertilization treatment and sampling period interaction (A × B) on average DHA (µg TPF 10 g^−1^ of air-dry soil) in Lessivated Cambisol under maize.

	Tested Parameters							
Ferilization treatment (A)	Control non-fertilized (Ø)	N1 (NPK 60:60:40)	N2 (NPK 120:120:120)	N1 + KP	N2 + KP	N1 + ES	N2 + ES	$\bar{X}$B
First sampling period (B)	67.86	66.72	22.69	90.62	32.22	97.26	34.52	58.84
Second sampling period (B)	77.79	74.79	24.66	96.12	39.06	111.66	40.72	66.40
Third sampling period (B)	71.42	69.86	23.72	99.16	34.62	102.12	35.89	62.40
$\bar{X}$A	72.36	70.46	23.69	95.30	35.30	103.68	37.04	62.55
Statistical analyses	Source of variation							
	Fertilization treatment (A)		Sampling period (B)		Fertilization treatment and sampling period interaction (A × B)			
LSD (0.05)	28.06 *		14.64 ^NSD^		48.70 ^NSD^			
LSD (0.01)	39.85 ^NSD^		20.81 ^NSD^		59.01 ^NSD^			

LSD: least significant differences at *p* = 0.05 and *p* = 0.01; * indicates statistical significant differences; ^NSD^ indicates no significant difference.

**Table 7 biology-09-00452-t007:** Effect of the fertilization treatment (A), sampling period (B), and fertilization treatment and sampling period interaction (A × B) on average PTA (number of gelatinolytic units g^−1^ of air-dry soil) in Lessivated Cambisol under maize.

	Tested Parameters							
Ferilization treatment (A)	Control non-fertilized (Ø)	N1 (NPK 60:60:40)	N2 (NPK 120:120:120)	N1 + KP	N2 + KP	N1 + ES	N2 + ES	$\bar{X}$B
First sampling period (B)	11.10	10.50	1.70	17.80	5.60	18.30	5.00	10.00
Second sampling period (B)	12.80	11.70	2.20	21.10	8.30	22.20	7.20	12.21
Third sampling period (B)	11.70	11.70	2.20	18.90	6.10	18.30	5.50	10.63
$\bar{X}$A	11.87	11.30	2.02	19.27	6.67	19.60	5.90	10.95
Statistical analyses	Source of variation							
	Fertilization treatment (A)		Sampling period (B)		Fertilization treatment and sampling period interaction (A × B)			
LSD (0.05)	3.11 ***		1.62 **		5.39 ^NSD^			
LSD (0.01)	4.14 ***		2.16 **		7.16 ^NSD^			

LSD: least significant differences at *p* = 0.05 and *p* = 0.01; ** and *** indicates statistical significant differences; ^NSD^ indicates no significant difference.

**Table 8 biology-09-00452-t008:** Effect of the fertilization treatments on yield and average chemical composition of the wheat grain (means ± standard deviation).

Fertilization Treatment (A)	Chemical Composition (%)				Yield (t ha^−1^)
	N	P	K	Proteins	
Ø	1.33 ^g^ ± 0.01	1.43 ^f^ ± 0.03	0.26 ^f^ ± 0.01	7.75 ^g^ ± 0.06	2.355 ^g^ ± 0.06
N1	1.51 ^f^ ± 0.02	1.56 ^e^ ± 0.01	0.43 ^e^ ± 0.02	8.80 ^f^ ± 0.03	4.998 ^f^ ± 0.04
N2	1.74 ^c^ ± 0.01	1.68 ^c^ ± 0.02	0.73 ^b^ ± 0.02	10.14 ^c^ ± 0.09	8.322 ^c^ ± 0.03
N1 + KP	1.61 ^d^ ± 0.03	1.59 ^d^ ± 0.04	0.54 ^d^ ± 0.03	9.39 ^d^ ± 0.03	6.133 ^e^ ± 0.01
N2 + KP	1.90 ^b^ ± 0.02	1.70 ^b^ ± 0.02	0.89 ^a^ ± 0.02	11.08 ^b^ ± 0.02	9.187 ^b^ ± 0.05
N1 + ES	1.60 ^e^ ± 0.03	1.60 ^d^ ± 0.01	0.57 ^c^ ± 0.04	9.33 ^e^ ± 0.01	6.265 ^d^ ± 0.02
N2 + ES	1.91 ^a^ ± 0.01	1.78 ^a^ ± 0.03	0.89 ^a^ ± 0.02	11.14 ^a^ ± 0.03	9.438 ^a^ ± 0.04
Statistical analyses	Source of variation				
	Fertilization treatment (A)				
P	***	***	***	***	***
LSD (0.05)	0.006	0.024	0.013	0.013	0.002
LSD (0.01)	0.008	0.033	0.018	0.017	0.003

LSD: least significant differences at *p* = 0.05 and *p* = 0.01; *** indicates statistical significant differences at the *p* < 0.05, *p* < 0.01 and *p* < 0.001 levels, respectively; Duncan’s Multiple Range Test (DMRT) was used to compare different variants at *p* ≤ 0.05, where values followed by the same letter in a column are not significantly different.

**Table 9 biology-09-00452-t009:** Effect of the fertilization treatments on yield and average chemical composition of the maize grain (means ± standard deviation).

Fertilization Treatment (A)	Chemical Composition (%)				Yield (t ha^−1^)
	N	P	K	Proteins	
Ø	1.24 ^e^ ± 0.03	0.87 ^f^ ± 0.01	0.22 ^e^ ± 0.04	7.75 ^e^ ± 0.08	1.952 ^f^ ± 0.03
N1	1.53 ^c^ ± 0.05	1.01 ^e^ ± 0.03	0.47 ^d^ ± 0.01	9.56 ^c^ ± 0.03	2.755 ^e^ ± 0.07
N2	1.65 ^b^ ± 0.01	1.19 ^c^ ± 0.07	0.70 ^b^ ± 0.02	10.31 ^b^ ± 0.01	3.789 ^b^ ± 0.09
N1 + KP	1.49 ^d^ ± 0.02	1.07 ^d^ ± 0.06	0.52 ^c^ ± 0.01	9.31 ^d^ ± 0.02	3.335 ^c^ ± 0.02
N2 + KP	1.69 ^a^ ± 0.06	1.23 ^a^ ± 0.02	0.90 ^a^ ± 0.06	10.56 ^a^ ± 0.05	4.133 ^a^ ± 0.01
N1 + ES	1.49 ^d^ ± 0.03	1.03 ^e^ ± 0.01	0.51 ^c^ ± 0.05	9.31 ^d^ ± 0.01	2.986 ^d^ ± 0.04
N2 + ES	1.66 ^b^ ± 0.01	1.21 ^b^ ± 0.05	0.91 ^a^ ± 0.07	10.38 ^b^ ± 0.02	4.133 ^a^ ± 0.03
Statistical analyses	Source of variation				
	Fertilization treatment (A)				
P	***	***	***	***	***
LSD (0.05)	0.016	0.004	0.018	0.019	0.044
LSD (0.01)	0.023	0.006	0.026	0.028	0.062

LSD—least significant differences at *p* = 0.05 and *p* = 0.01; *** indicates statistical significant differences at the *p* < 0.05, *p* < 0.01 and *p* < 0.001 levels, respectively; DMRT was used to compare different variants at *p* ≤ 0.05, where values followed by the same letter in a column are not significantly different.

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
