# Peer review of "Case Study upon Foliar Application of Biofertilizers Affecting Microbial Biomass and Enzyme Activity in Soil and Yield Related Properties of Maize and Wheat Grains"

_biology, 2020, doi:10.3390/biology9120452_

Round 1
Reviewer 1 Report
The resubmitted manuscript “Biofertilizers affecting microbial biomass ...” is better prepared than the first one. However, still, some corrections are needed.
General remarks.
The authors attribute the described effect of stimulating the yield of wheat and corn mainly due to the nitrogen fixation ability of tested strains KP and ES. But some strains of these bacterial species are well-known plant growth simulators due to the production of IAA and solubilization of mineral phosphates (e.g. DOI: 10.1088/1755-1315/308/1/012040; DOI: 10.1134/S0026261714050063; DOI: 10.21608/jsas.2017.1035.1004). These facts are practically not discussed in the manuscript.
Tables are still crowded and rearrangement of results will make them more comfortable for readers. I suggest rearranging the Tables like this;
|
Tested parameters |
Control non-fertilized |
N1 (NPK 60:40:40) |
N2 (NPK120:40:40) |
||||||
| O | KP | ES | MN1 | 0 | KP | ES | MN2 | ||
The total uptake by a grain of N, P, and K is still missed. If we compare the uptake of these nutrients by wheat and corn grain it is noticeable that the application of KP and ES increased at a similar level the total amount of up-taken P like N and noticeably more of K in comparison with NPK alone independently from the level of N fertilization. Such a comparison with respect to potential IAA secretion and P solubilization in the discussion will increase the value of this paper.
The effect of tested biofertilizers on the amount of uptaken N, P, and K by wheat and corn grain is also missed. If we compare the uptake of these nutrients by wheat and corn grain it is noticeable that the application of KP and ES increased at a similar level the total amount of up-taken P like N and noticeably more of K in comparison with NPK alone independently from the level of N fertilization. Such a comparison with respect to potential IAA secretion and P solubilization in the discussion will increase the value of this paper. Moreover, the interrelationships of the MBC and enzymatic activity with respect to the efficacy of NPK mixed with tested PGPB also are not discussed.
Author Response
Article:
Title
Case study upon foliar application of biofertilizers affecting microbial biomass and enzyme activity in soil and yield related properties of maize and wheat grains
Authors
Dragana Latković , Jelena Maksimović , Zoran Dinić , Radmila Pivić , Aleksandar Stanojković , Aleksandra Stanojković-Sebić *
Answers to Reviewer 1 report:
Details
Suggestion 1: The authors attribute the described effect of stimulating the yield of wheat and corn mainly due to the nitrogen fixation ability of tested strains KP and ES. But some strains of these bacterial species are well-known plant growth simulators due to the production of IAA and solubilization of mineral phosphates, which was practically not discussed in the manuscript.
Authors accepted reviewer suggestions; text changed with added description in the text; introduction is now more precised; according to improved text, authors inserted 3 references in introduction and totally rearranged numerical references.
Suggestion 2: Tables are still crowded and rearrangement of results will make them more comfortable for readers. Reviewer 1 suggests rearranging the Tables
Authors in general accepted reviewer suggestions; tables were changed and rearranged now from 2-7, instead tables 2 and 3, with additional description in the them; tables are now more precised (Line 164-179); tables 4 and 5 are now tables 8 and 9, bordered on all sides (Line 651 and 661)
Suggestion 3a: Discussion: The total uptake by a grain of N, P, and K is still missed. If we compare the uptake of these nutrients by wheat and corn grain it is noticeable that the application of KP and ES increased at a similar level the total amount of up-taken P like N and noticeably more of K in comparison with NPK alone independently from the level of N fertilization. Such a comparison with respect to potential IAA secretion and P solubilization in the discussion will increase the value of this paper.
Suggestion 3b: Discussion: The effect of tested biofertilizers on the amount of uptaken N, P, and K by wheat and corn grain is also missed. If we compare the uptake of these nutrients by wheat and corn grain it is noticeable that the application of KP and ES increased at a similar level the total amount of up-taken P like N and noticeably more of K in comparison with NPK alone independently from the level of N fertilization. Such a comparison with respect to potential IAA secretion and P solubilization in the discussion will increase the value of this paper.
Suggestion 3c: Discussion: The interrelationships of the MBC and enzymatic activity with respect to the efficacy of NPK mixedwith tested PGPB also are not discussed.
Authors accepted reviewer suggestions; text changed with added description in the text; discussion is now more precised; according to improved text, authors inserted 8 references in discussion and totally rearranged numerical references.

Reviewer 2 Report
The manuscript is now really improved and in appropriate quality for publication.
A few changes are needed
One term has to be changed, strictly: change soil surface into “upper soil”, in nearly all cases, a lot of cases being given though the whole text (not line 236) and in table 1 (Main chemical and physical properties of upper soil of Lessivated Cambisol) (in line 273: chang into soil materials) (line 321 data on soil surface chemical parameters to be changed in “data on chemical parameters from soil materials”)
Line 254: give the depth of soil sampllng and teh number of samples per plot.
Author Response
Article:
Title
Case study upon foliar application of biofertilizers affecting microbial biomass and enzyme activity in soil and yield related properties of maize and wheat grains
Authors
Dragana Latković , Jelena Maksimović , Zoran Dinić , Radmila Pivić , Aleksandar Stanojković , Aleksandra Stanojković-Sebić *
Answers to Reviewer 2 report:
Details
Suggestion 1: in nearly all cases term “soil surface” has to be change into “upper soil”.
Authors accepted reviewer suggestions, wording is corrected and „soil surface“ are replaced with “upper soil”
Suggestion 2: table 1 (Main chemical and physical properties of upper soil of Lessivated Cambisol), Lessivated Cambisol has to be change into “soil materials”.
Authors accepted reviewer suggestions, wording is corrected and in Table 1 „Lessivated Cambisol“ are replaced with “soil materials”
Suggestion 3: Line 321 in Reviewer 2 report: “data on soil surface chemical parameters” has to be change into “data on chemical parameters from soil materials”.
Authors accepted reviewer suggestions, wording is corrected and “data on soil surface chemical parameters” are replaced with “data on chemical parameters from soil materials” (Line 414 in our case study text)
Suggestion 4: Line 254 in Reviewer 2 report: give the depth of soil sampling and the number of samples per plot.
Authors accepted reviewer suggestion; additional explanation (the depth of soil sampling and the number of samples per plot) is added in the article (Line 333-334 in our case study text)

Reviewer 3 Report
In the manuscript entitled “Case study upon foliar application of biofertilizers affecting microbial biomass and enzyme activity in soil and yield related properties of maize and wheat grains”, the authors provides a field investigation of impact of microbial inoculation to crops yield and soil microbial activities. Overall, the questions addressed in the manuscript are well defined. The technique and results look at fine for me. However, the authors have chosen to combine results and discussion, which is not my preferred option to clearly distinguish the results from prior knowledge.
I have few comments on the text:
- The table 1 and 2 are very difficult to read. I prefer much the presentation of results in tables 3 and 4. After reading Table 1 and 2, I could not say easily which treatment gives significantly effect on the studied traits. Could the authors give a clear statistical comparison between the treatments for the table 1 and 2?
- Introduction: Some citations are omitted, line 60 to 71.
- Line 458: the sentence “Divers results can be found in various studies” sounds like tautologies. Please, be more precise.
- In the conclusion section, I prefer to read few take-home messages rather an exhaustive list of the results. Please, provide short conclusions in regard of the aims of the study.
Author Response
Article:
Title
Case study upon foliar application of biofertilizers affecting microbial biomass and enzyme activity in soil and yield related properties of maize and wheat grains
Authors
Dragana Latković , Jelena Maksimović , Zoran Dinić , Radmila Pivić , Aleksandar Stanojković , Aleksandra Stanojković-Sebić *
Answers to Reviewer 3 report:
Details
Suggestion 1: The table 1 and 2 are very difficult to read. Could the authors give a clear statistical comparison between the treatments for the table 1 and 2?
Authors in general accepted reviewer suggestions; tables were changed and rearranged now from 2-7, instead tables 2 and 3, with additional description in the them; tables are now more precised (Line 164-179); tables 4 and 5 are now tables 8 and 9, bordered on all sides (Line 651 and 661)
Suggestion 2: Introduction: Some citations are omitted, line 60 to 71.
Authors accepted reviewer suggestion; according to improved text, authors inserted 3 references in introduction and totally rearranged numerical references.
Suggestion 3: Line 416 (in Reviewer 3 comment), Line 695-698 (in our case study text): the sentence “Divers results can be found in various studies” sounds like tautologies. Please, be more precise.
Authors accepted reviewer suggestions, the term “Divers results can be found in various studies” is now changed and precised with added explanations in the text (Line 695-698)
Suggestion 4: In the conclusion section, provide short conclusions in regard of the aims of the study.
Authors accepted reviewer suggestions; text changed and conclusions are now precised, although the necessary conclusions should remain.

This manuscript is a resubmission of an earlier submission. The following is a list of the peer review reports and author responses from that submission.
Round 1
Reviewer 1 Report
Submitted manuscript “Biofertilizers affecting microbial biomass and enzyme activity in soil and yield related properties of maize and wheat grains” is interesting but is not new. The submitted manuscript described a well-known positive effect of the application of bacterial cultures on biological activity of soil and microbial communities.
However, the set-up of the experiment did not answer what was the main factor affecting the tested parameters of soil activity and the quality and quantity of the yield. either bacterial activity or nutrients of the media and metabolites of these bacteria.
The applied high doses of whole media with bacteria fertilized plots with a noticeable amount of proteins, amino acids, micro-nutrients, and probably of IAA. There have been a plethora of studies describing IAA-producing bacteria. The application of IAA-producing bacteria to plants has shown significant increases in plant growth and yield eg. Enterobacter ludwigii in rice [Nutaratat et al. 2017, 3 Biotech. 7(5):305] or in perennial ryegrass [Shoebitz et al 2009. Soil Biol Biochem. 41:1768) and Klebsiella pneumonia in wheat [Schadev et al. 2009, Ind. J. Exp. Biology, 47:993]. Moreover, the authors conclude eg. in line 432 "Excess microbiological fixed nitrogen ..." that was not supported by the appropriate measurement.
Summarizing, the manuscript is highly speculative in the parts related to the effect of tested microbial inoculants and to the explanation of their mode of action. Therefore, it can not be recommended for publication as it is.
Reviewer 2 Report
The authors seem not familiar with soil microbiology and soil ecology, despite tehe sentence in lines 103-105, with very or to general statement.
In general, soil microbial community harbors an immense diversity of strains, including strains favorable to plants. Inoculation with nitrogen fixing or PGPRB strains may be positive and is still usual in some regions, generally with soils having high sand part and less sorption capacity, also less contents of organic matter. Generally positive for sustainable soil fertility would be a crop rotation including legumes, that in general are deep rooting, with the development of N-fixers populations and soil structure amelioration.
Mineral fertilizers are not negative for microbiota in soils, when used in sense of sustainable agriculture without monoculture. The amounts should equalize the export by harvest, plus a part of losses, given also by meteorological events. Fertilizers enhance plant growth and development of plants biomass. Root biomass is enhanced as well as residues amounts at harvest. These higher amounts or organic compounds support the development and activity of microbiota in soil, not only microflora and microfauna, but also meso fauna with positive effects on soil structure, so humus formation with chemical stabilization of structure and pore formation.
So, the authors have to suppress the statements with only the negative statements on classic agriculture without direct input of microbial strains. The authors forget that nutrients exported by harvest have to be compensated by inputs of nutrients (However, not totally forgotten by the authors: A short sentence in text remember this fact). Nitrogen inputs may be ok by microbial fixers, esp. when being running the adequate crop rotation, P may be solubilized by bacteria from mineral compounds (also primary rocks etc). In general, degradation of minerals by inputs of organics from plants and biota deliver not enough amounts of nutrients to support a higher yield. Addition per fertilization are needed, organic fertilization being of course one possibility. Mineral fertilizing is not an obligation but allows better timing in course of plant development.
Details
Title: I propose “case study upon foliar application of biofertilizers affecting microbial biomass and enzyme activity in soil and yield related properties of maize and wheat grains“
Abstract:
Line 13 “2 rates of composite mineral fertilizers,“ in state of „different rates of composite mineral fertilizers,“
Line 30: eliminate „replacement“ or write “specify replacement of nitrogen fertilizers,”
details on analytical methods are not needed here in abstract.
Line 44 “application of the microbiological fixing bacteria“ what are „microbiological fixing bacteria“? Do you mean nitrogen fixing bacteria?
Lines 50-52: You have to specify that these consequences occur only in case of extreme mismanagement, not in case of best practice.
Line 94 “indicates a higher respiration potential“ in state of „indicates a higher respiration rate“
Fig.1 Title: Temperatures and precipitations developments for wheat and maize vegetation period in 2012/2013 and 2014, respectively, for the field study locality.
Line 154: what is an associative strain? Your strains are from collections, grown under conditions far away from soil ecology. You must remember that in the discussion, by validation of effects of the strains.
2.3. Bacterial inoculation. You inoculate plant biomass/surface and soil surface, never the soil. Give precision about the distribution: parts on plants, parts on soil surface (you say: soil inoculation!). It seems the inoculated bacteria have no contact to the plant roots. Have they enough access to organic matter to be able to reduce/fixe atmospheric nitrogen? This process it highly energy consuming and bacteria start this activity only in case of N-deficiency. You must consider this fact by the validation of plant grain quality.
Lines 203 etc: Do you eliminate the plant roots from the soil material or is the rhizosphere part of the analyzed soil material? You must give the precision and you must consider the effects of rhizosphere on MBC DHA etc in case root were co-analysed.
Table 2, foot note “microbial biomass carbon, in mg kg-1 of an absolutely dry soil“ - explain this unit, as you had only air drying before analysis, and you precise air-dry for DAH and PTA. In case absolutely dry is ok: give precision on drying in material & methods, and the reason for analysis of MBC in absolutely dry soil material.
Table 3: you present only results for 2 sampling times in maize, (line 206 „and three times during the vegetation period of maize (intensive plant growth stage, 7-8 leaves - sampling period I, milk-waxy maturity stage - sampling period II, full grain maturity stage - sampling period III)“. Why not results for all samplings. Give at minimum the sampling time for sample B (second or third sampling)
Tables 4-5: it is ok to give Chemical composition of grains, p.ex for validation of quality as food or feed. For validation of yield and of the effects of treatments you must give also the total exports of N P K by harvest for the treatments +. (no problem as yield is given for each treatment), and you must discuss this result.
Line 298: you forgot the role of microfauna + mesofauna, regulating the microbial populations and communities, and preparing organic residues for microbial attack, p.ex. . I propose you specify the role of microfauna fro microbial life in soils.
Line 367, Line 432. You postulate: “The excess of microbiological fixed nitrogen, obtained by increased activity of applied microbial inoculants,“ . Have you data about the nitrogen fixation? About the increased activity? DHA and PTA indicate not the actual activity but the potential activity. You analyzed never N-fixation-activity. I expect you had not N-fixation by bacteria spread on soil surface (see need of energy, of organic matter; need of reduction of oxygen concentration, microaerophilic) and on leaves. To postulate nitrogen fixation is here not allowed considering your experimental approach and the analysis. To postulate excess of microbiological fixed nitrogen is far away from scientific soundness! Your willingness to prove positive effects of inoculated strain is not of scientific quality! The scientific community is well informed about the fact, that PGPB are an important parameter of plant productivity. It is not necessary to find new arguments on basis on wrong facts. In your case I would expect other effects of bacteria on plants (PGPB on leaves). I refer also on lines 429, suggesting no N-fixation as bacteria are well supplied with nitrogen in this soil after “long-term application of mineral fertilizers to the soil at high rates“.
The statement in line 429 has to be given in Chapter “2.1. Field trial and weather conditions“ by the description of the experimental site. Data on total N content in Table 1 suggest however not a special high nitrogen content. It seems, statement in line 429 on “this study” is wrong.
Line 390:
The conclusion sentence with the information “indicated that the type of soil,….“ is not allowed as you had no approach with the comparison of different soil types at different sites. This first sentence in 4.Conclusion must be eliminated. The second sentence give your main information: wheather conditions were globally more important for yield than an addition of bacteria, in your case with conditions favorable in 2013, and wet in 2014 (lines 130-133). The determining/limiting parameter in your field experiment overcomed management inputs in 2014.
Line 435: new:
„It was found that the combined application of microbial inoculants and lower doses of mineral NPK fertilizers, increased the yield of maize and wheat compared to the use of only NPK fertilizers in our case study. An extrapolation to other sites is not possible, given especially the meteorological conditions in 2014
A general remark for the discussions: a lot of works cited were realized in pots and in green house. It is better to specify the pot-experiments, as generally realized under optimized conditions, and with results not to be generalized.
I propose the editor to accept the manuscript for publication, however only after revision.
Reviewer 3 Report
The authors present a field study on microbial inoculation to two different crops to quantify the impact on nutrient synthesis in situ versus nutrient amendment (externally), and the impacts on microbial growth and activity, as well as crop biomass. The issue is not the study, it is the presentation of the data. It took quite a while to figure out what was going on in this manuscript, because the presentation of the data and the text is so disorganized. It is actually unacceptable to present data in a manner this convoluted. However, the actual study is interesting and relevant - so it is not a matter of the data, but how the authors choose to revise the data presentation so a reader does not have to struggle just to figure out what is going on.
Table 2: this is crowded and unreadable, and the fact that we have to look in the footnotes to find the applicable units is un acceptable. put the units in the legend, or as part of the actual table with +/- STDEV. Also, split the statistical analyses out of this table and make them another table, or just describe them in the text. it took me a lot of time to make sense of these data.
Table 3: exact same comments as table 2
Table 4: what do the lowercase letters mean? this was not defined anywhere and it detracts from the data.
Table 5: exact same comments as table 4
Text: I acknowledge that Serbian to English translation mist be difficult as would any ESL translation. This one reads as if Google translate was used. Please find a native English writer that also is fluent in Serbian, and make sure it is completely revised. As an example, in the first sentence the phrase "which tends to high yields" likely means "which favors higher yields" is lost in translation (literally...)
There is good work here that is obscured by bad presentation. If the authors can rewrite and re-present the data in a clear manner, then it is more reasonable as a publication.